# MRI-Based Radiomics Differentiates Skull Base Chordoma and Chondrosarcoma: A Preliminary Study

**DOI:** 10.3390/cancers14133264

**Published:** 2022-07-03

**Authors:** Erika Yamazawa, Satoshi Takahashi, Masahiro Shin, Shota Tanaka, Wataru Takahashi, Takahiro Nakamoto, Yuichi Suzuki, Hirokazu Takami, Nobuhito Saito

**Affiliations:** 1Department of Neurosurgery, Graduate School of Medicine, The University of Tokyo, 7-3-1 Hongo, Bunkyo-ku, Tokyo 113-8655, Japan; ekondou-ryk@umin.ac.jp (E.Y.); takami-tky@umin.ac.jp (H.T.); nsaito-nsu@m.u-tokyo.ac.jp (N.S.); 2RIKEN Center for Advanced Intelligence Project, 2-1 Hirosawa, Wako 351-0198, Japan; satoshi.takahashi.fy@riken.jp; 3Division of Medical AI Research and Development, National Cancer Center, 5-1-1 Tsukiji, Chuo-ku, Tokyo 104-0045, Japan; 4Department of Neurosurgery, University of Teikyo Hospital, 2-11-1 Kaga, Itabashi-Ku, Tokyo 173-8606, Japan; 5Department of Radiology, The University of Tokyo Hospital, 7-3-1 Hongo, Bunkyo-ku, Tokyo 113-8655, Japan; wataru.harry1@gmail.com (W.T.); tnakamoto@hs.hokudai.ac.jp (T.N.); suzukiy-rad@h.u-tokyo.ac.jp (Y.S.); 6Department of Biological Science and Engineering, Faculty of Health Sciences, Hokkaido University Kita 12, Nishi 5, Kita-ku, Sapporo-shi 060-0808, Japan

**Keywords:** chondrosarcoma, chordoma, diagnostic performance, machine learning model, MRI, radiomics

## Abstract

**Simple Summary:**

In this study, we created a novel MRI-based machine learning model to differentiate skull base chordoma and chondrosarcoma with multiparametric signatures. While these tumors share common radiographic characteristics, clinical behavior is distinct. Therefore, distinguishing these tumors before initial surgical intervention would be useful, potentially impacting the surgical strategy. Although there are some limitations, such as the risk of overfitting and the lack of an extramural cohort for truly independent final validation, our machine learning model distinguishing chordoma from chondrosarcoma yielded superior diagnostic accuracy to that achieved by 20 board-certified neurosurgeons.

**Abstract:**

Chordoma and chondrosarcoma share common radiographic characteristics yet are distinct clinically. A radiomic machine learning model differentiating these tumors preoperatively would help plan surgery. MR images were acquired from 57 consecutive patients with chordoma (N = 32) or chondrosarcoma (N = 25) treated at the University of Tokyo Hospital between September 2012 and February 2020. Preoperative T1-weighted images with gadolinium enhancement (GdT1) and T2-weighted images were analyzed. Datasets from the first 47 cases were used for model creation, and those from the subsequent 10 cases were used for validation. Feature extraction was performed semi-automatically, and 2438 features were obtained per image sequence. Machine learning models with logistic regression and a support vector machine were created. The model with the highest accuracy incorporated seven features extracted from GdT1 in the logistic regression. The average area under the curve was 0.93 ± 0.06, and accuracy was 0.90 (9/10) in the validation dataset. The same validation dataset was assessed by 20 board-certified neurosurgeons. Diagnostic accuracy ranged from 0.50 to 0.80 (median 0.60, 95% confidence interval 0.60 ± 0.06%), which was inferior to that of the machine learning model (*p* = 0.03), although there are some limitations, such as the risk of overfitting and the lack of an extramural cohort for truly independent final validation. In summary, we created a novel MRI-based machine learning model to differentiate skull base chordoma and chondrosarcoma from multiparametric signatures.

## 1. Introduction

Chordoma is a rare midline skull base tumor, with an incidence of approximately 8.4 per ten million per year [1]. It is predominantly found in the skull base and clival area and is considered to originate from the notochordal remnants. Chondrosarcoma of the skull base is even rarer, with an incidence of less than 2 per ten million per year [2]. It is found in the petroclival region and spheno-ethmoidal sinus, originating from the synchondroses of the petro-occipital sutures or the spheno-ethmoidal sutures. Chordoma and chondrosarcoma are the two most common primary bone tumors of the skull; other tumors, including plasmacytoma, myeloma, and metastasis, are rarely seen in clinical practice [3]. Chordoma and chondrosarcoma share common radiographic characteristics on computed tomography and magnetic resonance images (MRI), although the clinical behavior of these tumors is distinct. Chordoma is characterized by an aggressive nature and has a high risk of repeated recurrence despite multimodal treatments, including extensive surgical resection and high-dose radiotherapy [4]. In contrast, chondrosarcoma shows a benign clinical phenotype, with a high vulnerability to radiotherapy leading to a better prognosis than chordoma [5]. While the 10-year overall survival (OS) of chordoma is 32.3% [6], that of chondrosarcoma ranges from 89.9% to 100% [7]. Due to the inconsistent efficacy of radiotherapy, including heavy particle radiotherapy and high-dose radiosurgery in chordoma, extensive surgical resection of the tumor and adjacent bony structures is recommended [8]. Furthermore, radiotherapy is indicated not only for residual or recurrent tumors but also for the resection cavity after gross total resection for prophylactic purposes [9]. In contrast, in chondrosarcoma, a “wait-and-scan” approach may be clinically justified not only after gross total resection but also after subtotal resection [10]; as radiotherapy is expected to be highly effective, a judicious approach aiming at safe maximum resection avoiding critical anatomical structures such as internal carotid arteries at the surgery would be practical [11]. In view of their distinct clinical behavior, it is critical to distinguish these tumors that appear to be radiographical “twins” for human eyes prior to initial surgical intervention. This information would impact the timing of surgery, the choice of surgical approach, and the extent of resection. However, no reliable method for distinguishing chordoma and chondrosarcoma preoperatively is currently available.

The skull base forms the floor of the cranial cavity and separates the brain from other facial structures. The anterior skull base proper is formed medially by the cribriform plate, which forms the roof of the nasal cavity; laterally, by the orbital plates of the frontal bone, which form the roof of the orbits and ethmoid air cells; and posteriorly, by the planum sphenoidale and lesser wings of the sphenoid. Percutaneous image-guided biopsy for definitive histologic diagnosis was achieved in 86% (12/14) of cases in the skull [12] and 81.3% (13/16) of cases in the craniovertebral junction [13]. However, for ‘central’ skull base tumors, image-guided biopsy is not safe due to the presence of the vital adjacent structures described above. A novel, non-invasive, and reliable diagnostic method is desirable.

Radiomics is an emerging and rapidly developing approach to medical imaging, which enhances the existing data by extracting a large number of features from the images and detects disease characteristics by applying mathematical analyses [14]. MRI-based radiomics has been developed for various diseases [4,15,16,17,18,19,20,21]. To the best of our knowledge, only one study has reported the efficacy of a radiomic approach to distinguish chordoma from chondrosarcoma in the skull base region [2]. Radiomic analysis was performed based on a strictly regulated data acquisition protocol using the same MRI machine for all cases, with outstanding diagnostic performance. However, overwhelming specification regarding imaging acquisition and modality hampers the versatility and generalization of the diagnostic algorithm, limiting its clinical use. Thus, this study aimed to develop a machine learning model to distinguish skull base chordoma and chondrosarcoma that is potentially applicable to a broad range of diseases and for use in multicenter collaborative studies.

## 2. Materials and Methods

### 2.1. Cases and Magnetic Resonance Imaging (MRI) Sequences

MR images were acquired for consecutive patients with chordoma or chondrosarcoma surgically treated at The University of Tokyo Hospital between September 2012 and February 2020. This study was approved by the ethics committee of the University of Tokyo (IRB no. 11770).

Cases satisfying the following three criteria were eligible for the study:(1)Two types of MR images were acquired for the preoperative evaluation: T2-weighted images (T2) and T1-weighted images with gadolinium enhancement (GdT1).(2)No history of surgical resection of skull base lesions.(3)All images were acquired on imaging systems with field strengths of 1.5 T or 3.0 T and different vendors. The imaging parameters for each system are shown in Appendix A.

The study included 57 consecutive patients (32 chordomas and 25 chondrosarcomas, Table 1). The first consecutive 47 datasets from 47 patients (27 chordoma and 20 chondrosarcoma; September 2012 to October 2018) were used to develop the model (“training dataset”). The last 10 datasets from 10 patients (5 chordoma and 5 chondrosarcoma; October 2018 to February 2020) were used for independent testing of the diagnostic performance (“final validation dataset”). All the diagnoses were determined histopathologically after surgery.

### 2.2. Feature Extraction

Volumes of interest (VOIs) were manually delineated on the GdT1 and T2 images for all cases using a radiation treatment planning system (Monaco^®^ ver. 5.11, Elekta, Stockholm, Sweden) by a board-certified neurosurgeon (E.Y.).

An outline of the feature extraction is shown schematically in Figure 1. The radiomic features consisted of morphological, intensity, histogram, and texture features. First, 8 morphological features were extracted from VOI binary masks (e.g., volume, sphericity, etc.). Next, a 3-dimensional wavelet transformation was applied to all MR images. Coiflet 1, which is often used as a mother wavelet transformation, was applied to each of the volume axes (*x*, *y*, and *z*), resulting in 8 wavelet-transformed images (LLL, HLL, LHL, LLH, HHL, HLH, LHH, and HHH) in addition to the original image, where L represents low frequency and H represents high frequency (Appendix A). From each of these 9 images, 18 intensity features such as maximum intensity and minimum intensity and 20 histogram features such as maximum histogram gradient intensity were extracted. The number of bins for histogram features was fixed to 64 for quantization [22]. The texture features were also extracted from the 9 images. The number of bins for quantization was refined with 16, 32, 64, and 128 for each image before calculating the texture features [23,24]. The texture features consisted of 11 gray-level co-occurrence matrix (GLCM), 13 gray-level run-length matrix (GLRLM), 13 gray-level size-zone matrix (GLSZM), 16 neighboring gray-level dependence matrix (NGLDM), and 5 neighborhood gray-tone difference matrix (NGTDM) (IBSItextbook_v9: https://arxiv.org/abs/1612.07003, accessed on 1 June 2019). In total, 8 + 9 × (20 + 18) + 9 × 4 × (11 + 13 + 13 + 16 + 5) = 2438 features were obtained per image sequence (Appendix A). Therefore, the number of features per case was 2438 × 2 = 4876, where 2 represents the number of image sequences used (T2 and GdT1).

The above feature extraction was performed semi-automatically using a MATLAB programming tool for radiomic analysis (2019a; Math Works, Natick, MA, USA) [23]. More details on the feature extraction methods are described in our previous study [18].

For the description of a feature, bin number and characters represent the type of MR images (T2 and GdT1) followed by the type of wavelet transformation and the type of feature. For example, *6bit_GD_HHL_GLCM_ Inverse difference moment* represents the inverse difference moment feature extracted from HHL wavelet-decomposed 64bin GdT1. The value of each feature was standardized using the *z*-score normalization method.

### 2.3. Machine Learning Model Creation

We created and evaluated the models in three major phases “*Feature selection Phase*”, “*Model*
*selection Phase*”, and “*Test Phase*”. A schematic figure is illustrated in the Results.

### 2.4. Feature Selection Phase

First, we determined the most appropriate feature number to create a machine learning model by simulation of the training dataset using logistic regression-recursive feature elimination (LG-RFE). In total, 47 cases (27 chordoma and 20 chondrosarcoma) attending our institution between September 2012 and October 2018 were used (“training dataset”). The number of features was set from 1 to 10 based on the medical literature [25]. LG-RFE created a model from features and then deleted one feature with the lowest importance sequentially by measuring the contribution of each feature in a given model. This process was repeated until one feature remained. Cases in the training dataset were randomly divided into 4 subgroups; 3 of these were used for training, and the remaining subgroup was used for feature number validation. The above processes were repeated 5 times, and the mean accuracy was calculated. The feature number in which the highest mean accuracy was achieved was finally identified (Appendix A).

Next, features were selected by LG-RFE. This time RFE started by creating a model from all the 2431 features, and the process was repeated until the predetermined number of features remained. This was repeated 5 times, and the most frequently selected features were chosen. For feature selection, a pairwise correlation coefficient was calculated in each imaging sequence. A pair of feature values with a pairwise correlation coefficient exceeding 0.7 were considered to contain redundant information.

### 2.5. Model Selection Phase

Parameters were set using Grid Search with nested 2-fold cross-validation within the training dataset, using 80% of the 47 cases in the training dataset. Prediction models with the selected features and parameters were created with two representative machine learning methods: logistic regression (LR) and support vector machine (SVM).

Evaluation of the learning models was performed based on the average area under the curve (AUC) by the following procedure:(1)The training dataset was divided randomly into 5.(2)Four out of five datasets were used for training, and the remaining dataset was used for testing.(3)A receiver operating characteristic (ROC) curve was drawn to obtain the AUC.(4)Steps 2–3 were repeated 5 times to obtain the average AUC.

### 2.6. Test Phase (Final Validation)

The latest 10 datasets from 10 patients (5 chordoma and 5 chondrosarcoma; October 2018 to February 2020) were used for the independent testing of the diagnosis performance for each machine learning model (“final validation dataset”).

### 2.7. Diagnostic Performance: Machine Learning Model vs. Neurosurgeons

MR images of the above final validation dataset were evaluated by 20 board-certified neurosurgeons who were not involved in the treatment of the 10 patients participating in this trial. The diagnostic performance of the neurosurgeons to distinguish chordomas and chondrosarcomas on the MR images was compared to that of the machine learning models. The difference in diagnostic accuracy between the neurosurgeons and the machine learning model was assessed, and details of each case were discussed.

### 2.8. Statistical Analyses

Python 3.8.10 was used for statistical analyses (https://www.python.org/downloads/release/python-3810/) (accessed on 3 June 2021).

The Python packages used were NumPy, pandas, math, and sklearn (scikit-learn: https://scikit-learn.org/stable/) (accessed on 3 June 2021).

To compare the diagnostic accuracy between the neurosurgeons and the machine learning model, the one-sample *t*-test was used [26], and a *p* value less than 0.05 was considered significant.

## 3. Results

### 3.1. Machine Learning Models

Eight features from GdT1 features and two features from T2 features were selected. A pair of the selected feature exhibited a high correlation coefficient (>0.7) and was deemed redundant. We removed one feature to make all correlation coefficients below 0.7. As a result, seven features from GdT1 features and two features from T2 features were selected (Table 2).

We selected the best parameters using nested two-fold cross-validation for each SVM and LG (Figure 2). When using the seven features extracted from GdT1, the AUC was 0.93 ± 0.06, and the accuracy in the final validation dataset was 0.90 (9/10) in the logistic regression model. The AUC was 0.89 ± 0.10, and the accuracy was 0.70 (7/10) in the SVM model. When using the two features extracted from T2, the AUC was 0.87 ± 0.07, and the accuracy in the final validation dataset was 0.50 (5/10) in the logistic regression model. The AUC was 0.86 ± 0.05, and the accuracy was 0.40 (4/10) in the SVM model. When using both the seven features extracted from GdT1 and the two features extracted from T2, the AUC was 0.95 ± 0.07 and accuracy was 0.70 (7/10) in the logistic regression model, and the AUC was 0.92 ± 0.07 and accuracy was 0.70 (7/10) in the SVM model **(**Figure 3, Table 3).

In summary, the machine learning model that distinguished chordoma from chondrosarcoma with the highest accuracy was the model created using the seven features extracted from GdT1 and logistic regression.

### 3.2. Diagnostic Performance: Machine Learning Model vs. Human Neurosurgeons

Diagnosis by the 20 neurosurgeons was performed in the 10 cases in the final validation dataset. The percentage of neurosurgeons with a correct diagnosis varied between 25% and 85%. The diagnostic accuracy, which represents the percentage of correct answers by each surgeon, varied between 40% and 80% among the neurosurgeons (median 60%, 95% confidence interval: 60.5 ± 6.4%).

In the best-performing machine learning model, the diagnostic accuracy was 90% (9/10) which was significantly better than that of the neurosurgeons (*p* = 0.031). Of note, the Shapiro–Wilk test (*p* = 0.07) suggested that the data were normally distributed. The machine learning model correctly diagnosed the nine cases, while the diagnostic accuracy of the neurosurgeons in these cases varied between 40% and 85% (median 65%). The machine learning model was incorrect in one case of chondrosarcoma, in which the diagnostic accuracy of the neurosurgeons was 25%.

Representative cases of chordoma and chondrosarcoma are presented in Figure 4.

## 4. Discussion

Radiomics has been studied for the purpose of preoperative evaluation of histological diagnosis and tumor consistency in various neoplasms such as prostate cancer [16], lung cancer [18], and brain tumors including glioma [17,19,20,21], meningioma [27], and brain metastasis [28]. The radiographic characteristics are extracted as parametric data, and a large number of radiomic features are analyzed together with clinical variables [29]. Radiologists and clinicians commonly make a diagnosis using “visible” radiographic features and tend to rely on their knowledge and experience. Radiomic analyses employ different approaches; the extracted features are mathematically calculated from digital images, most of which are “invisible” to the human eye [30,31]. This study aimed to distinguish skull base chordoma and chondrosarcoma based on MRI. We assessed various patterns of multiparametric signatures and found that the prediction model adopting seven features extracted from GdT1 in the logistic regression revealed the best performance in the final validation dataset. The results of this study suggest that MRI-based radiomic methods may be useful to radiologists and neurosurgeons to distinguish these two “radiographically similar but clinically different” skull base tumors and develop an appropriate surgical strategy.

Percutaneous CT-guided biopsy for definitive histologic diagnosis is a golden standard for tumors in accessible skeletal sites such as the spine and appendicular skeleton. However, for ‘central’ skull base tumors, CT-guided biopsy is not safe due to the presence of surrounding vital structures.

The difficulty in differentiating these two tumors on MRI was revealed in a test performed on 20 board-certified neurosurgeons. Generally, tumor laterality contributes to the preoperative prediction by neurosurgeons. Cases that saw high percentages of correct answers tended to be located at the typical tumor location: centrally for chordoma and laterally for chondrosarcoma (Figure 4A–F). The chordoma case with the lowest neurosurgeon diagnostic accuracy (40%) while correctly diagnosed by the machine learning model was located slightly laterally (Figure 4G–I). The only case that the machine learning model misdiagnosed was centrally located chondrosarcoma, and the diagnostic accuracy of the neurosurgeons was also low (20%) (Figure 4J–L). Of note, the current machine learning model did not take the tumor location into account. The aim of our study was to develop a versatile, purely radiomic model independent of experienced neurosurgeons. Furthermore, the above cases suggest that adding the information on tumor location would not necessarily improve the diagnostic accuracy of the machine learning model.

Only one previously published study has reported a machine learning model discriminating chordoma and chondrosarcoma in the skull base [2]. This model was established based on the T1, T2, and GdT1 images obtained from 210 cases. Although the number of cases is significantly greater than that in this study, the diagnostic accuracy in the validation dataset was lower (72.85%) than this study’s final validation dataset (90%). Furthermore, the diagnostic accuracy using only T1-derived features was not satisfactory (AUC 0.55), and yet, the proposed model incorporated four T1-derived features. This may, in part, explain the difference in diagnostic accuracy between the above study and this study that utilizes features derived from GdT1 but not T2 and T1.

Other than machine learning models, the differences in MRI characteristics between skull base chordoma and chondrosarcoma have been explored in the previous decade [32,33,34]. One study suggested that the different distribution of apparent diffusion coefficient values on the diffusion-weighted image (DWI) may be useful. DWI is readily available without any special equipment; however, it may not be suitable for radiomic analysis given its insufficient resolution.

Radiomics is not only used in diagnosing chordoma but also in predicting the control rate of chordoma with treatment [35,36]. The model built on radiomic features describing tumor shape and disomic heterogeneity achieved the best performance in predicting tumor control with radiotherapy [35]. In view of the refractory nature of the disease requiring multiple surgical procedures and radiotherapy, a radiomic model predictive of sustained tumor control would be useful in the treatment of chordoma.

The features selected are shown in Table 2. Four GLSZM features, two intensity features, two GLRLM features, and one GLCM feature were selected. Most of the features are mathematically calculated from digital images and “invisible” to the human eye; their relationship with the clinical characteristics of chordoma and chondrosarcoma is yet to be determined. Nonetheless, this radiomic model used the radiomic features of unknown clinical significance and successfully distinguished these two tumors. In our view, this highlights the significance of radiomics. Although 4 GLSZM features were selected, the IBSI textbook notes that the predictive performance of GLSZM-based features or their reducibility has not been determined. The prediction model adopting seven features extracted from GdT1 in the logistic regression disclosed the best performance in the final validation dataset. This finding may suggest that features extracted from GdT1 are more useful in distinguishing chordoma and chondrosarcoma.

Some limitations of this study should be noted. First, the study enrolled a relatively small number of cases (27 chordomas and 20 chondrosarcomas in the training dataset: 5 chordomas and 5 chondrosarcomas in the final validation dataset). When using a small dataset to create a machine learning model, overfitting is one of the greatest concerns. In our study, when using the two features extracted from T2, the AUC was 0.87 ± 0.07 and the accuracy in the final validation dataset was 0.5 (5/10) by the logistic regression model, and the AUC was 0.86 ± 0.05, and the accuracy in the final validation dataset was 0.4 (4/10) by the support vector machine model, which may suggest overfitting. With our relatively small datasets, it is hard to completely avoid overfitting, but we adopted two preventive measures to avoid this as far as possible: we used semiautomatic decision in the number of features using RFE and semiautomatic selection of features using RFE; we also used the final validation dataset. Second, this study lacked an extramural cohort for truly independent final validation. Therefore, further research is warranted. Ideally, studies should be prospective with a data acquisition protocol shared by different study institutions. Nonetheless, the results of this study demonstrate that machine learning models could achieve an acceptable accuracy even with a relatively small dataset and basic pulse sequences such as T2 and GdT1, which may offer a platform for multicenter collaborative studies [37]. Third, a detailed evaluation of the stability of features was not performed. In this study, we focused on the feasibility of classification between chordoma and chondrosarcoma using a relatively small MR image dataset. Building a classification model that is sufficiently robust to feature variability across scanners will require further research.

Future directions of this study include radiogenomic analysis incorporating the genetic profiles of these tumor entities. Studies have suggested that tyrosine kinase receptors, including platelet-derived growth factor receptor and epithelial growth factor receptor and their downstream pathway of phosphoinositide 3-kinases, Akt, and mammalian target of rapamycin are aberrantly activated in chordoma [38,39,40]. Multiple molecularly targeted agents have been or are currently being tested for chordoma with promising results (NCT03083678, NCT04042597) [4,41,42,43]. In the future, reliable, radiogenomic diagnosis may allow us to treat a locally aggressive chordoma with a neoadjuvant small molecule and lessen the serious risks of surgery that ensues. Segmentation of the tumor was performed manually in this study. If deep learning could eliminate this potential human bias by automated segmentation, our model would be more robust and readily applicable to other radiomic studies of brain tumors in the future.

## 5. Conclusions

In this study, we developed a novel MRI-based radiomic model that distinguishes skull base chordoma and chondrosarcoma with excellent diagnostic accuracy. This model will assist neurosurgeons in differentiating these two skull base tumors with similar radiographic characteristics but different clinical behavior and in developing the appropriate surgical approach in individual cases.

## Figures and Tables

**Figure 1 cancers-14-03264-f001:**
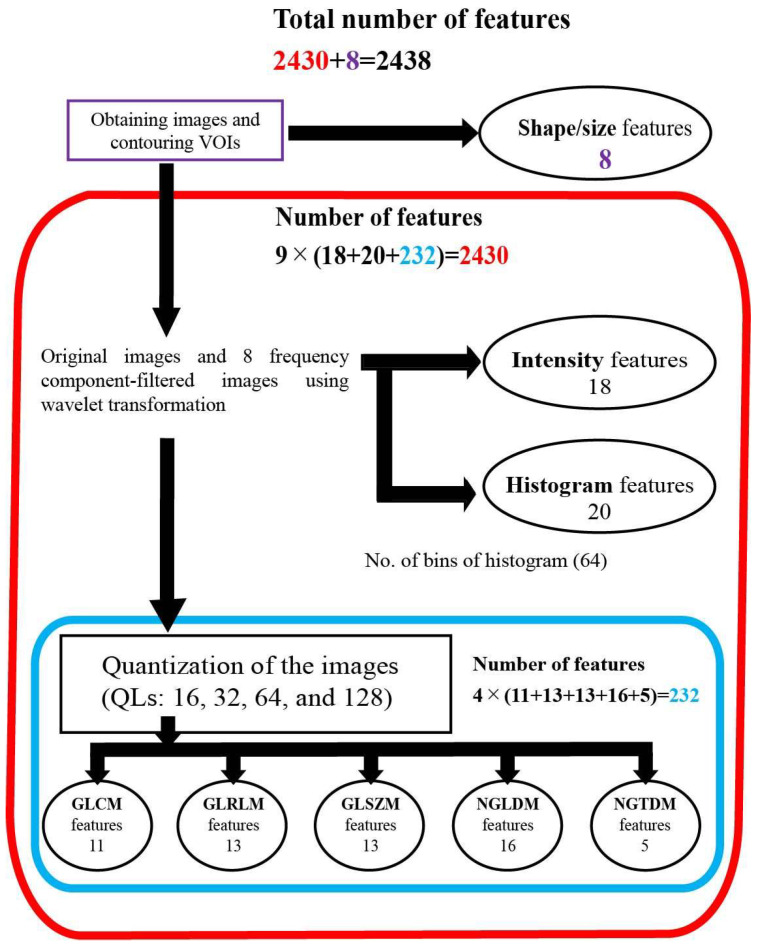
Diagram of features extraction. A total of 2438 features were extracted from each MRI sequence (GdT1 and T2). Abbreviations: quantization levels—QLs; gray-level co-occurrence matrix—GLCM; gray-level run-length matrix—GLRLM; gray-level size-zone matrix—GLSZM; neighboring gray-level dependence matrix—NGLDM; neighborhood gray-tone difference matrix—NGTDM.

**Figure 2 cancers-14-03264-f002:**
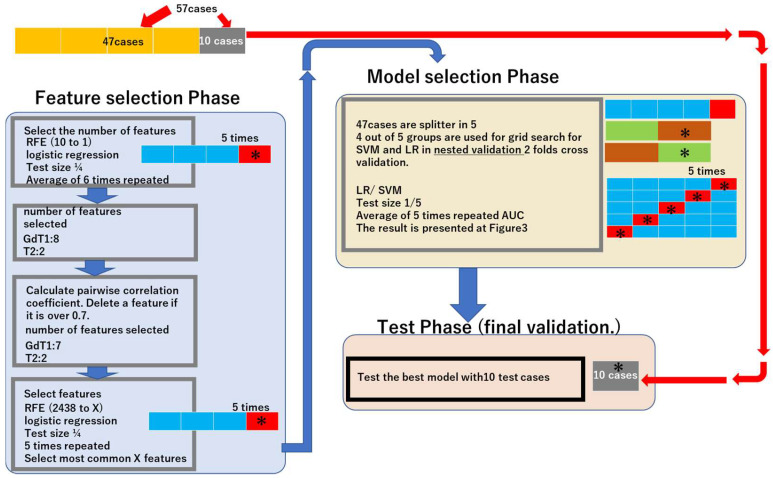
Workflow. The workflow is composed of 3 main phases, “Feature selection Phase”, “Model selection Phase”, and “Test Phase”. ” Feature selection Phase” includes determination of appropriate feature number and feature selection. “Model selection Phase” includes parameter determination with nested validation and evaluation of the learning model. “Test Phase” is a final test of models with 10 independent cases for final validation. * Test cases. X: 7 for GdT1, 2 for T2. Abbreviations: recursive feature elimination—RFE; area under the curve—AUC; logistic regression—LR; support vector machine—SVM.

**Figure 3 cancers-14-03264-f003:**
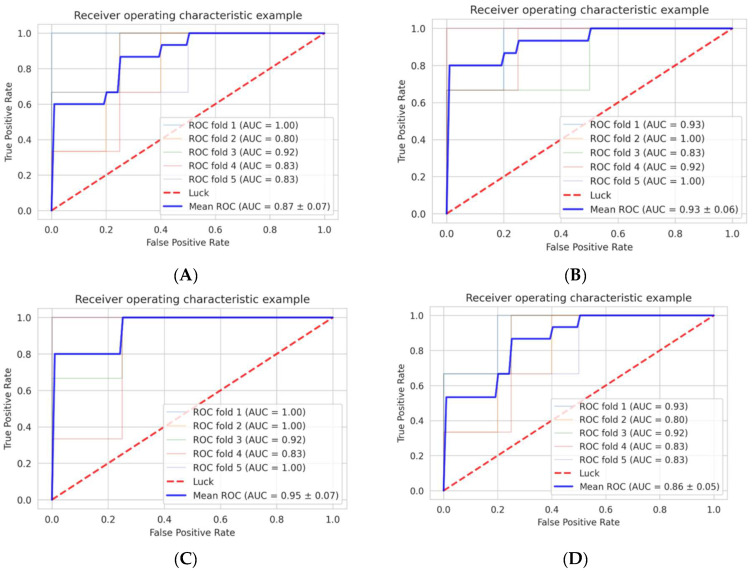
Diagnostic accuracy of the machine learning models in the training dataset (refer to Figure 2). (**A**) The AUC of the model with T2 logistic regression. (**B**) The AUC of the model with GdT1 logistic regression. (**C**) The AUC of the model with T2 and GdT1 logistic regression. (**D**) The AUC of the model with T2 SVM. (**E**) The AUC of the model with GdT1 SVM. (**F**) The AUC of the model with T2 and GdT1 SVM. The number of each mean ROC represents mean +/− standard deviation of the 6 AUCs. Abbreviations: receiver operating characteristics—ROC; area under the curve—AUC; T2 weighted image—T2; post-gadolinium T1-weighted images—GdT1; support vector machine—SVM.

**Figure 4 cancers-14-03264-f004:**
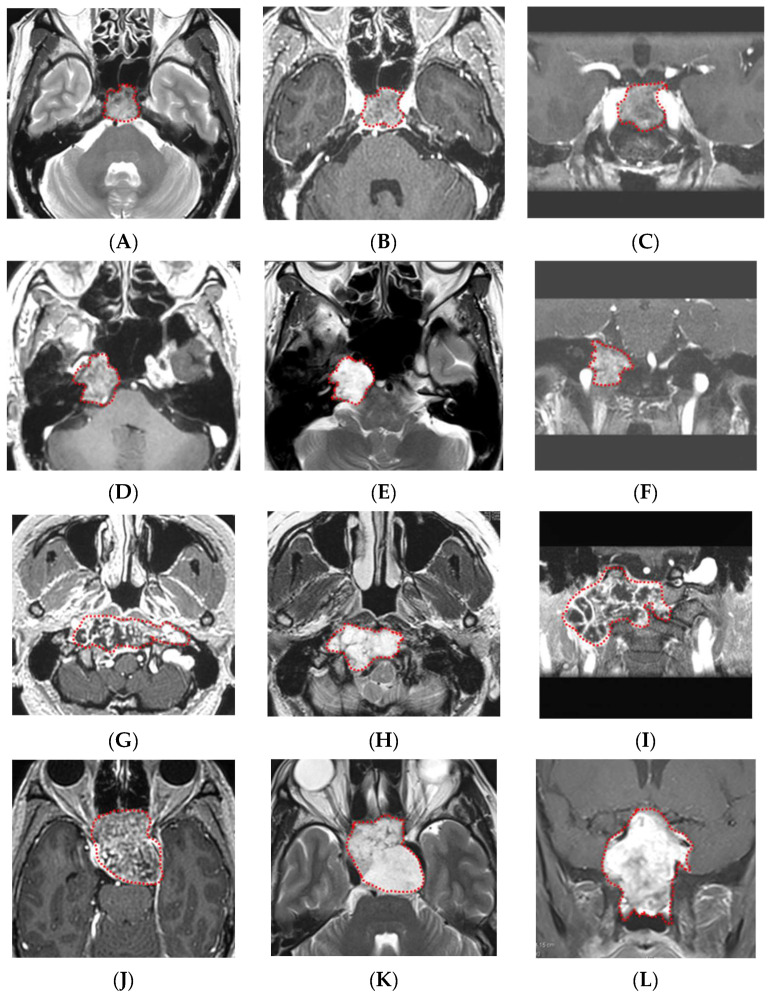
Representative cases of chordomas and chondrosarcomas. (**A**–**C**) GdT1, T2, and fsGdT1 coronal images of a typical chordoma case. The machine learning model correctly diagnosed the tumor, and high diagnostic accuracy (80%) by neurosurgeons was noted. (**D**–**F**) GdT1, T2, and fsGdT1 coronal images of a typical chondrosarcoma case. The machine learning model correctly diagnosed the tumor, and the highest diagnostic accuracy by the neurosurgeons (85%) was noted. (**G**–**I**) GdT1, T2, and fsGdT1 coronal images of the chordoma case with the correct diagnosis by the machine learning model but with low diagnostic accuracy by the neurosurgeons (40%). (**J**–**L**) GdT1, T2, and fsGdT1 coronal images of the chondrosarcoma case with the incorrect diagnosis by the best machine learning model and the lowest diagnostic accuracy by the neurosurgeons (20%). Abbreviations: T2 weighted image—T2; post-gadolinium T1-weighted images—GdT1; fat suppression post-gadolinium T1-weighted images—fsGdT1. Red dotted lines indicates tumor.

**Table 1 cancers-14-03264-t001:** Patient and tumor characteristics.

	Training Dataset		Final Validation Dataset	
	Chordoma	Chondorosarcoma	Chordoma	Chondrosarcoma
Case number	27	20	5	5
Median age	51.0	40.5	57.5	64.0
Sex				
Male	15	11	4	0
Female	12	9	1	5
Median volume (cm^3^)	12.7	9.5	6.1	12.0

**Table 2 cancers-14-03264-t002:** Selected features for machine learning models.

MRI Sequence	Wavelet	Quantization	Type	Name	Frequency
GdT1	HLL	-	Intensity	Mean	2
	HHL	4-bit (16)	GLCM	Correlation	2
	HLH	4-bit (16)	GLSZM	High gray-level zone emphasis (HGZE)	1
	LLL	4-bit (16)	GLSZM	Zone size variance (ZSV)	1
	LLL	5-bit (32)	GLSZM	Low gray-level zone emphasis (LGZE)	1
	LLH	7-bit (128)	GLRLM	Gray-level variance (GLV)	1
	LHH	-	Intensity	Skewness	1
T2	LHL	4-bit (16)	GLSZM	Gray-level non-uniformity (GLN)	2
	HHH	4-bit (16)	GLRLM	Gray-level non-uniformity (GLN)	1

**Table 3 cancers-14-03264-t003:** AUC and diagnostic accuracy of machine learning models.

	Logistic Regression	Support Vector Machine
T2	0.87 ± 0.07 (0.4)	0.86 ± 0.05 (0.4)
GdT1	0.93 ± 0.06 (0.9) *	0.89 ± 0.10 (0.7)
T2 + GdT1	0.95 ± 0.07 (0.7)	0.92 ± 0.07 (0.7)

The numbers in each cell represent AUC (mean ± standard deviation) and diagnostic accuracy in parentheses. Abbreviations: area under the curve—AUC; T2-weighted image—T2; post-gadolinium T1-weighted image—GdT1. * Highest accuracy.

## Data Availability

Not applicable.

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
