# Peer review of "MRI-Based Radiomics Differentiates Skull Base Chordoma and Chondrosarcoma: A Preliminary Study"

_cancers, 2022, doi:10.3390/cancers14133264_

Round 1

Reviewer 1 Report

Thank You for the work done.

Author Response

Dear Reviewer,

Thank you very much for your time in reviewing our manuscript.

We are very glad to hear your approval.

Sincerely,

Shota Tanaka

Reviewer 2 Report

I read both the manuscript and the response 2 of rejection.   All raised comments have been answered.   In addition to the comments made: For me is unclear, whether these patients have had  a histopathological diagnosis?  I could not find an answer, it is only mentioned that preoperative patients are included. I guess, the radiological features were compared later to the  correct histopathological diagnosis?  This should be added.      I would also  propose to add that  these are preliminary data, not tested on different machines and  should be validated.  Maybe even in the title the word preliminary should be added  or that this concerns a pilot study. 

Author Response

Dear Reviewer,

Thank you very much for the review.

We have answered your questions and comments point-by-point as seen below.

Sincerely,

Shota Tanaka

In addition to the comments made: For me is unclear, whether these patients have had a histopathological diagnosis?  I could not find an answer, it is only mentioned that preoperative patients are included. I guess, the radiological features were compared later to the correct histopathological diagnosis?  This should be added.     

Per your suggestion, we have added the sentence in p.3 l.129.

“All the diagnoses were determined histopathologically after surgery.”

 I would also propose to add that these are preliminary data, not tested on different machines and should be validated.  Maybe even in the title the word preliminary should be added or that this concerns a pilot study.  

Per your suggestion, we have changed our title as below:

“MRI-based radiomics differentiates skull base chordoma and chondrosarcoma: A preliminary study”

Also, we would like to let you know that, following the editor’s suggestion, we have added the limitations of this study in Abstract and Simple Summary.

“Although there are some limitations such as the risk of overfitting and the lack of an extramural cohort for truly independent final validation.”

Round 2

Reviewer 2 Report

All comments have been adressed sufficiently

This manuscript is a resubmission of an earlier submission. The following is a list of the peer review reports and author responses from that submission.

Round 1

Reviewer 1 Report

I am satisfied with the revisions performed,

Thank You for the work done.

I would only suggest You to add in the Introduction the rate (%) of bone neoplasms occurring within this peculiar region (do a sort of list). Are chondrosarcoma and chordoma the two most common?
Myeloma, and metastases are rarer or easier to distinguish from chordoma and chondrosarcoma on conventional imaging? Please discuss it briefly.

This would perfectly finally create the frame for this really nice and novel paper.

Author Response

Dear Reviewer,

We thank you for taking the time to review our manuscript.

We also appreciate your positive feedback. We have made some amendments according to your suggestion (highlighted in red in the revised manuscript).

We believe that these revisions have improved the quality of our manuscript and hope that it now meets the standard for publication in your journal.

Sincerely,

Shota Tanaka

I would only suggest you to add in the Introduction the rate (%) of bone neoplasms occurring within this peculiar region (do a sort of list). Are chondrosarcoma and chordoma the two most common? Myeloma, and metastases are rarer or easier to distinguish from chordoma and chondrosarcoma on conventional imaging? Please discuss it briefly.

Response: Per your suggestion, we have addressed the etiology of primary bone tumors of the skull in the Introduction. As you suspect, chordoma and condrosarcoma are the two most common primary bone tumors of the skull, although there are other, rarer tumors including plasmacytosis, myeloma, and metastasis. Please understand that, because of the limitation in the total number of figures and tables, we would comment on the list of primary bone tumors of the skull in the main text. We have also added a new reference (Kekkar A, et al. J Neurol Surg B Skull Base 2016).

Reviewer 2 Report

The authors tried to assess my comments, but I still have some concerns that the authors must address before publication. Also, I would suggest authors to be more precise in the revision of the manuscript. I have the feeling that the first round of revision was coarse, which is not admissible when presenting a manuscript to a journal such as Cancers. Again, the clarity of the manuscript is limited, so I would suggest author to pay attention on this as readers have to properly understand what you did. I also strongly suggest the revision of English language.

  1. Introduction:
    • You added a section in the introduction (line 84-92). Please clarify why you added this. If it is because you want to highlight that biopsy is difficult in deep-seated chordoma and thus you are proposing radiomics, please specify explicitly. In the actual form is not clear.
    • Please, also clarify because in this new section you are mentioning about CT-guided biopsy, when your analysis is limited to MRI. Why not including CT in your analysis?
  2. M&M:
    • You added Table 1, but in the text you are still reporting that this table is in supplementary material. Please correct. Honestly, I would leave table 1 in supplementary.
    • I understand that the data are limited and you couldn’t check about features robustness among different scanners. However, I believe that an analysis of features stability is mandatory for this work. You should account for features stability at least by applying known perturbation (e.g. stability wrt noise, rotation, translation, and similar).
    • Section 2.4.2 – what is NA ? please clarify. I suppose NaN values, why you have these values ?
    • In section 2.4.2 it is still not clear how many featues have been selected ? 10 as reported in the previous paragraph? Please, clarify. I would probably merge section 2.4.1 with section 2.4.2
    • Section 2.5. Please specify what is 80% of the training set. Do you mean 80% of the 27 data you used?
    • Line 205 – test phase. I can’t understand if this is a paragraph or not.
    • Statistical analysis. Is your distribution normal to perform a t-test ?
  3. Results
    • Line 227. What do you mean with “We removed one feature to make all correlation coefficient below 0.7.” ?
    • Table 3 report the frequency of the selected feature – the frequency is very low. How did you computed it? If a feature compares just once, how you define that this is stable ?
    • You should move Figure 2, in the method section.
    • Are the results of Table 4 relevant to cross-validation or testing ?
    • Figure 4. Please highlight the contour of the chordoma.
  4. Discussion:
    • Sentence at line 325: “The aim of our study was to 325 create a versatile, purely radiomic model readily applicable to any institutions independ-326 ent of experienced neurosurgeons.” This is too strong. Before applying your model to other institutions, you should test it internally and externally.
    • Lines 351-364: This section is a repetition of the results. My suggestion was to comment the selected features from a clinical point of view. i.e. what does it means clinically that a specific feature was selected ? you should link the mathematical formula with a clinical meaning, considering that you are evaluating the model also in terms of clinical decision-making.
